# Tutorial for Stopped-Flow Water Flux Measurements: Why a Report about “Ultrafast Water Permeation through Nanochannels with a Densely Fluorous Interior Surface” Is Flawed

**DOI:** 10.3390/biom13030431

**Published:** 2023-02-24

**Authors:** Juergen Pfeffermann, Peter Pohl

**Affiliations:** Institute of Biophysics, Johannes Kepler University, Gruberstraße 40, 4020 Linz, Austria

**Keywords:** water channel, aquaporins, hydrodynamics, membranes, light scattering, lipid vesicles

## Abstract

Millions of years of evolution have produced proteinaceous water channels (aquaporins) that combine perfect selectivity with a transport rate at the edge of the diffusion limit. However, Itoh et al. recently claimed in *Science* that artificial channels are 100 times faster and almost as selective. The published deflation kinetics of vesicles containing channels or channel elements indicate otherwise, since they do not demonstrate the facilitation of water transport. In an illustrated tutorial on the experimental basis of stopped-flow measurements, we point out flaws in data processing. In contrast to the assumption voiced in *Science*, individual vesicles cannot simultaneously shrink with two different kinetics. Moreover, vesicle deflation within the dead time of the instrument cannot be detected. Since flawed reports of ultrafast water channels in *Science* are not a one-hit-wonder as evidenced by a 2018 commentary by Horner and Pohl in *Science*, we further discuss the achievable limits of single-channel water permeability. After analyzing (i) diffusion limits for permeation through narrow channels and (ii) hydrodynamics in the surrounding reservoirs, we conclude that it is unlikely to fundamentally exceed the evolutionarily optimized water-channeling performance of the fastest aquaporins while maintaining near-perfect selectivity.

## 1. Introduction

Water scarcity is a global problem expected to become increasingly pressing in the future, as no globally affordable technical solutions for water separation have yet been identified. Mimicking the water-selective membranes of biological systems may appeal as the desired solution: in every kingdom of life, nature employs very narrow channels that are only about one water molecule wide, chief among them aquaporins. Such geometry enables perfect selectivity [1] because hydrated ions are too bulky to pass [2].

Unfortunately, hitherto aquaporin-containing membranes for water separation represent neither an affordable nor scalable solution to the problem of water scarcity. Among many reasons, the comparatively thick aquaporin wall renders the ratio of the cross-sectional area of the pore lumen to the total area occupied by the channel unfavorable for technical applications [3]. Nanofiltration membranes from aquaporins might only reach good porosities if the channels were packed in 2D crystals, which thus far appears unattainable on industrial scales. Widening the channel lumen above 3–4 Å would also increase porosity. Yet, natural channels with such a lumen are no longer water selective [4,5,6].

Consideration of this background motivates the search for artificial water channels. In 2017, a report in Science claimed to have found the solution: extremely narrow carbon nanotubes [7]. Due to their thin walls and extremely high unitary water permeability, *p*_f_, these single-walled tubes appeared to be an excellent solution. Yet, their reported *p*_f_—surpassing the fastest aquaporins—came along with exceedingly high activation energy, *E*_A_, for water flow across these tubes. The combination of these two observations constituted a problem [8]. The well-known Arrhenius equation predicts that *E*_A_ and *p*_f_ are correlated. *E*_A_ reflects the energetic barrier to water transport: if *E*_A_ is large, *p*_f_ is small, and vice versa (Figure 1A). An exception would only be possible if a gain in entropy compensated for the high expense in enthalpy. Yet, we found no significant entropy difference between intraluminal and bulk water molecules [9]. Accordingly, transition state theory allows calculating *p*_f_ from *E*_A_ and vice versa [8,9]. In conclusion, the carbon nanotube study could not provide sound experimental evidence that very narrow carbon nanotubes are superior to nature’s aquaporins.

In 2022, Science published another promising report: a team of Japanese researchers synthesized fluorous oligoamide nanorings, which were supposed to polymerize in phospholipid bilayer membranes to form nanochannels [10]. Given its 0.9-nm wide channel lumen, the artificial ^F12^NC_4_ channel showed an astonishing preference for water over ions. Moreover, with a reported *p*_f_ of 5.5 × 10^−10^ cm^3^ s^−1^, water allegedly crossed this nanochannel at a 100 times higher rate than the fastest aquaporin (GlpF [11]).

Unfortunately, upon inspection, we realized that instead of miraculously efficient channels, flawed analysis of the underlying stopped-flow data yielded those high *p*_f_ values. The experimental data, which became available only months after the original publication and after repeated requests, do not support the claim regarding rapid ^F12^NC_4_-facilitated water transport, whereby ^F12^NC_4_ has the reportedly highest *p*_f_ among the supramolecular nanochannels presented by Itoh et al. [10]. This conclusion, drawn from our reassessment of the experimental data, comes as no surprise, because the extreme *p*_f_ falsely attributed to the 0.9-nm wide ^F12^NC_4_ channel would have violated the limits set by diffusion across narrow channels. Moreover, additional consideration of hydrodynamics outside the channel shows that the adjacent aqueous solution could not feed ^F12^NC_4_ with enough water molecules to allow for such rapid water conductance.

We approach the reassessment of the experimental data by first outlining the fundamental idea behind water flux measurements in vesicular systems using the stopped-flow technique, namely the monitoring of water flux across the vesicular membrane after introducing an osmotic gradient. As we will later discuss data from both stopped-flow light scattering and fluorescence self-quenching experiments, we introduce these approaches at reading out the change in intravesicular volume due to transmembrane water flux. Throughout, we note several critical requirements for accurate determinations of *p*_f_. Eventually, in the Results section, we present our interpretation of the experimental data from both experimental approaches that Itoh et al. [10] intended to affirm the exceptional water-channeling properties of ^F12^NC_4_ and demonstrate where the authors went wrong. Finally, the Discussion is dedicated to a theoretical analysis of what water permeabilities narrow water channels may achieve, which concludes that aquaporins may not be surpassed by much.

## 2. A Brief Tutorial on Water Flux Measurements Using Stopped-Flow (Materials and Methods)

### 2.1. Vesicles in an Osmotic Gradient

Large unilamellar vesicles (LUVs) built from self-assembling amphiphilic lipids—herein simply referred to as vesicles—represent a convenient means for assessing unitary channel permeability, *p*_f_. Under appropriate conditions, the liposomes behave like perfect osmometers [12,13]. They increase or decrease their volume under the influence of osmotic gradients. The changes in volume can be assessed in real-time by monitoring (i) the intensity of light scattered by the LUVs [14] or (ii) the concentration-dependent quenching of fluorophores encapsulated in the LUVs [15]. The stopped-flow technique ensures the rapid exchange of extravesicular osmolarity, which effectively synchronizes the individual vesicle shrinkage kinetics and allows for monitoring fast changes in volume (Figure 2).

The rate of change of intravesicular volume, d*V*(*t*)/d*t*, upon exposing a vesicle with initial intravesicular osmolyte concentration *c*_0_ to hyperosmotic conditions reflects the overall volumetric flux of water, *J*_v_ (in units of cm^3^/s), across the vesicular membrane. Here, we are concerned with the case where the vesicular membrane, consisting of the lipid bilayer and any embedded water-conducting channels, is not permeable to the respective osmolyte in the time frame relevant for the stopped-flow experiment. Consequentially, the number of intravesicular osmolyte particles remains constant, yet their concentration depends on *V*(*t*). Upon denoting the incremental osmolyte concentration in the extravesicular volume as *c*_s_, we can write for the osmotic gradient Δ*c*(*t*):(1)$$\Delta c\left(t\right)=\frac{V_{0}}{V\left(t\right)}c_{0}-\left(c_{0}+c_{s}\right)$$
where *V*_0_ is the intravesicular volume at time zero. *J*_v_ may comprise a relevant contribution from (a) the bare lipid bilayer, *J*_v,l_, which, depending on the lipid composition, can be substantial, and (b) any membrane-embedded water-conducting channels, whereby we refer to the channel-awarded increment in volumetric flux as *J*_v,c_. Experimentally, one obtains *J*_v_ and is subsequently tasked with determining *J*_v,l_ and *J*_v,c_, whereby *J*_v_ = *J*_v,l_ + *J*_v,c_.

*J*_v,l_, the volumetric flux of water across the lipid bilayer, is given by the following expression:(2)$$J_{v,l}=P_{f,l}V_{w}A_{l}\Delta c\left(t\right)$$
where *P*_f,l_ (in units of cm/s) is the lipid bilayer’s osmotic permeability coefficient, *V*_w_ the molar volume of water, and *A*_l_ the vesicular surface area occupied by the lipid bilayer. *J*_v,c_, the total volumetric flux of water through all membrane-embedded water-conducting channels, is defined as follows [16]:(3)$$J_{v,c}=p_{f}V_{w}n\Delta c\left(t\right)$$
where *p*_f_ (in units of cm^3^/s) is the channel’s unitary water permeability coefficient. Since *J*_v,l_ and *J*_v,c_ add up, we can write:(4)$$J_{v}=J_{v,l}+J_{v,c}=\left(P_{f,l}A_{l}+p_{f}n\right)V_{w}\Delta c\left(t\right).$$

For practicality, we choose to define *P*_f,c_, which reflects the contribution of membrane-embedded water-conducting channels to the experimentally observed vesicle shrinkage kinetics, as [16,17]:(5)$$P_{f,c}=p_{f}n/A_{v}$$

While *A*_l_ may deviate from the vesicle’s surface area *A*_v_ in case a sizeable fraction of the membrane was occupied by channels, in the experiments we are concerned with here, this is not the case. Itoh et al. indicated *n* values well below 10 and an outer diameter between 5.2 nm and 6.3 nm for their fluorous nanochannels. Even if 10 of their channels were reconstituted into a 100-nm diameter vesicle, *A*_l_ would differ from *A*_v_ by <1%. This assumption of a negligible difference between *A*_v_ and *A*_l_ is commonly also made for vesicles containing reconstituted protein channels, since the external diameter of many water-conducting channels is in the range of a few nanometers, e.g., a typical aquaporin monomer measures roughly 2.5 nm, the respective tetramer somewhat more than twice that [18] and *n* seldomly exceeds 20. If it does, the background permeability of the lipid is much smaller than the combined water permeability of the channels so that *J*_v,l_ << *J*_v,c_, i.e., *J*_v_ = *J*_v,c_ [19]. Thus, we approximate *A*_l_ with *A*_v_, which is experimentally accessible, e.g., using dynamic light scattering.

Dividing both sides of Equation (4) by *A*_l_ = *A*_v_ yields:(6)$$J_{v}/A_{v}=\left(P_{f,l}+p_{f}n/A_{v}\right)V_{w}\Delta c\left(t\right)$$

Rearranging the above equation and inserting Equation (5) yields:(7)$$P_{f,l}+P_{f,c}=\frac{J_{v}}{A_{v}V_{w}\Delta c\left(t\right)}=P_{f}$$

Upon substitution of Equations (1) and (5) into Equation (4), we can rewrite the latter as:(8)$$\frac{dV\left(t\right)}{dt}=A_{v}P_{f}V_{w}\left(\frac{V_{0}}{V\left(t\right)}c_{0}^{}-\left(c_{0}^{}+c_{s}\right)\right)$$

Equation (8) describes the time course of osmotically-induced vesicle volume changes. We want to highlight that the additivity of *P*_f,l_ and *P*_f,c_ is limited to cases where *A*_l_ ≈ *A*_v_. Equation (8) has the analytical solution [14]:(9)$$V\left(t\right)=V_{0}\frac{c_{0}^{}}{c_{0}^{}+c_{s}}\left\{1+L\left(\frac{c_{s}}{c_{0}^{}}\exp\left(\frac{c_{s}}{c_{0}^{}}-\frac{A_{v}P_{f}V_{w}{\left(c_{0}^{}+c_{s}\right)}^{2}}{V_{0}c_{0}^{}}t\right)\right)\right\}$$
where *L* is the Lambert function, defined as $L\left(x\right)e^{L\left(x\right)}=x$.

Itoh et al. [10] assumed a homogenous distribution of channels between vesicles. This assumption is reasonable since the hydrophobic nanorings which supposedly formed the supramolecular nanochannels were already added to the lipids prior to vesicle formation. Variable channel densities per vesicle are often found upon reconstitution of natural protein channels into lipid bilayers [20] (Figure 2).

### 2.2. Assessing Vesicle Deflation Kinetics in a Liposome Suspension by Measuring the Intensity of Scattered Light

*V*(*t*) is experimentally accessible by measuring the intensity of light scattered by the vesicles *I*(*t*) (Figure 3). The Rayleigh–Gans–Debye relation describes the dependence of *I*(*t*) on vesicle radius *R* [21]:(10)$$I∼\left(\frac{\lambda^{2}}{4\pi^{2}}\right){\left(\frac{m^{2}-1}{m^{2}+2}\right)}^{2}\delta^{6}\left(\frac{1+\cos^{2}\theta }{2}\right)P\left(\theta \right)$$
where *λ*, *θ*, and *m* are the effective wavelength (i.e., the ratio of the wavelength of incident light *λ*_0_ and the refractive index *n*_s_ of the aqueous solution), the angle at which the intensity *I* of scattered light is measured, and the relative refractive index (*m* = *n*_p_/*n*_s_, where *n*_p_ is the average refractive index of the particle). *R* enters the equation via the size parameter δ=2πR/λ and the form factor $P\left(\theta \right)={\left(3\left(\sin u-u\cos u\right)/u^{3}\right)}^{2}$, where $u=2\delta \sin\left(\theta /2\right)$.

Using the explicit expression for *I* is rather tedious [14]. Indeed, its Taylor expansion provides a sufficient level of accuracy:(11)$$I\left(t\right)=a+bV\left(t\right)+dV^{2}\left(t\right)$$

Although it is possible to derive the coefficients *b* and *d* from the solution of Equation (10), they are commonly found by fitting a second-degree polynomial (Equation (11)) to *I*(*t*) [14].

### 2.3. Evaluating Vesicle Deflation Kinetics in a Liposome Suspension by Measuring the Fluorescence Intensity of Encapsulated Aqueous Dyes

Self-quenching of aqueous dyes is distance-, and thus, concentration-dependent. This observation can be used to follow the upconcentration of a dye during vesicle deflation (Figure 4). Since vesicles behave like perfect osmometers, a linear dependence between fluorescent light intensity, *I*_f_(*t*), and vesicle volume is attainable. This requires a sufficiently high fluorophore concentration. For the frequently used carboxyfluorescein, the initial intravesicular concentration should be above 10 mM [13,22].

## 3. Results

### 3.1. Reassessment of Experimental Data Obtained in Stopped-Flow Light Scattering Experiments with ^F12^NC_4_

We focused our data evaluation on the narrowest described channel, ^F12^NC_4_, built from ^F12^NR_4_ rings [10]. Since the original data became available only months after the original paper appeared, we digitized the published stopped-flow light scattering curve (Figure 5B, red points) and used Equation (9) to infer *P*_f_. After the raw data had become publicly available, we repeated the same fitting procedure. To our surprise, the published graph and the raw data do not match (Figure 5B,C, red points).

For the published plot of ^F12^NC_4_ channels in DOPC (dioleoyl phosphatidylcholine) vesicles, we find *P*_f_ = 69 µm/s (Figure 5B, black line). This value corresponds to the permeability of a bare DOPC lipid bilayer [23,24], and hence, *P*_f_ = *P*_f,l_. It follows that *P*_f,c_ = 0 (or *P*_f,c_ << *P*_f,l_), which agrees with the observation that the time course of scattered light intensity shown in ref. [10] for bare DOPC liposomes is practically identical to that in the presence of ^F12^NC_4_. If *P*_f,c_ was equal to 0.25 cm/s, as claimed in [10], we would expect the measured scattering trace (Figure 5B, red points) to resemble the blue trace in Figure 5B—yet, there is no resemblance.

The analysis of the raw data (https://doi.org/10.5061/dryad.h18931zq1) yields a different picture (Figure 5C, red points): two apparently different deflation kinetics indicate sample inhomogeneity, i.e., the presence of two distinct vesicle populations. Accordingly, we adopted the fitting algorithm described in the caption to Figure 5. We can assign the major fraction *f*_1_ = 64% to channel-free DOPC vesicles since *P*_f,1_ = 61 µm/s (Figure 5C). The origin of the minor fraction (*f*_2_ = 36%) with fast kinetic is unclear. It may originate from vesicle lysis upon osmotic challenge, aggregates in the sample, or other shortages in sample preparation or device application. In any case, ^F12^NC_4_ channel activity cannot be involved. In disregard of the apparent inability of an individual vesicle to simultaneously shrink with two distinct time constants, Itoh et al. proceeded to calculate *P*_f,c_ and *P*_f,l_ from the two time constants extracted by a two-exponent fit. Figure 2D illustrates under which conditions such kinetics may be observable, given the assumption of a homogeneous channel-containing vesicle ensemble: all channels must have been concertedly blocked shortly after the onset of the osmotic challenge, so that water may pass afterward solely through the lipid bilayer, utterly undisturbed by any channel activity.

A more realistic scenario that could explain the experimental data is illustrated in Figure 2C. It specifies that only a fraction of vesicles may contain ^F12^NC_4_ channels. Since channel-containing and empty liposomes deflate at different rates, the scattering trace obtained from the inhomogeneous vesicle ensemble will exhibit two distinct kinetics. Unfortunately, the experimental data presented by Itoh et al. [10] render the applicability of this scenario in their experiments unwarranted. It would require (a) independent proof that only ≈1/3 of the vesicles contain the functionally-assembled channel and/or (b) deflation experiments where the functional channel is present in most vesicles. Yet, Itoh et al. indicated that on average, about 1.9 channels should be active per vesicle for the experiment shown in Figure 5. We calculated that number from the mentioned % mol fraction of 0.0053 ^F12^NR_4_, the number of 3.5 × 10^5^ lipids per vesicle, and the involvement of 10 ^F12^NR_4_ molecules per channel. We computed the number of lipids from the vesicle diameter of 200 nm, a lipid area of 0.72 nm^2^ [25], and the presence of 2 leaflets per vesicle. The data leave no room for empty vesicles, thereby ruling out (a). Furthermore, Itoh et al. provided experimental data in which they gradually increased the concentration of ^F12^NR_4_. Yet, the fraction with fast kinetic, *f*_2_, did not increase, rendering option (b) obsolete as well (Figure 6).

### 3.2. Reassessment of Experimental Data Obtained in Stopped-Flow Fluorescence Self-Quenching Experiments with ^F12^NC_4_

Eventually, Itoh et al. [10] postulated that DOPC is poorly suited for studying water permeation through their nanochannels due to “spontaneous water leakage through the fluidic vesicular membrane”. The argument is incorrect because the extreme *p*_f_ values reported for their nanochannels should render *P*_f,l_ of DOPC negligible (Section 3.1). However, Itoh et al. felt that *p*_f_ of their supramolecular channels is better assessable in DPPC (dipalmitoyl phosphatidylcholine) liposomes since *P*_f,l_ of DPPC in its gel phase is nearly two orders of magnitude lower than DOPC’s *P*_f,l_ in the fluid phase [26].

With radii as small as ~24 nm, as read from the published raw data (https://doi.org/10.5061/dryad.h18931zq1), the DPPC vesicles were much smaller than the DOPC vesicles. From the % mol fraction of 0.0018 indicated for ^F12^NR_4_ (Figure 7), we find 0.54 ^F12^NR_4_ molecules per vesicle. Channel formation appears very unlikely, since (i) ^F12^NR_4_ distribution between the vesicles should be homogenous and (ii) ten ^F12^NR_4_ molecules would be required per channel. We base our conclusion on the finding of only 3 × 10^4^ lipids per vesicle, as computed from the lipid area of 0.479 nm^2^ [27] and the presence of 2 leaflets per vesicle.

Despite these problematic settings, Itoh et al. [10] reported channel activity in DPPC vesicles. Instead of measuring light scattering, they now assessed *V*(*t*) via the fluorescence intensity of encapsulated carboxyfluorescein (Figure 7). Within 10 ms, as reported in their Science paper, bare DPPC vesicles do not shrink. Only channel-containing vesicles may respond to the osmotic challenge. The “representative” curve in the Science paper (reproduced in Figure 7) may falsely suggest that such vesicles exist, despite the above estimate of only 0.054 channels per vesicle. However, a glimpse at how the curve was obtained leads to a different conclusion.

In contrast to the exemplary curve in the Science paper, which displays the fluorescence intensity in the interval between 2 and 10 ms, Figure 7A shows the same trace starting from 0 ms. Fluorescence intensity increases in the interval between 0 and 2 ms—in contrast to the expected fluorescence decrease due to self-quenching upon vesicle shrinkage. As an explanation, Itoh et al. [10] offered that they used a “monochromator with a measured dead time of 1.1 ms”. We do not know why a monochromator should have a dead time. Yet, even if we attributed this dead time to sample mixing, we would still be left with a 0.9 ms interval of unexplained fluorescence increase. Since Figure 7A averages several stopped flow shots, we hoped to find an explanation by looking at the individual curves (Figure 7B). Only one out of eight traces exhibits a fluorescence decrease starting with the end of the dead time. The seemingly stochastic pattern of these traces raises the question of whether averaging is warranted.

To illustrate how fluorescence intensity should have changed given the reported experimental parameters, we calculated the expected volume change from the published value for *P*_f,c_ and used the result to compute the corresponding relative fluorescence intensity (Figure 7C). Thus, we found that the vesicles should have been largely deflated within the 1.1 ms dead time of the stopped-flow device and even more so within the first 2 ms. Thus, the very design of the stopped-flow self-quenching experiment does not allow for concluding that ^F12^NC_4_ facilitates water transport at the reported rate.

## 4. Discussion

The ultrafast water channels surpassing aquaporins’ *p*_f_ by a factor of 100 while retaining excellent selectivity do not exist. The study of Itoh et al. [10] did not provide compulsory experimental evidence. It relied on ill-designed stopped-flow approaches. The DPPC experiments were flawed for at least two reasons: (i) the vesicles contained only trace amounts of ^F12^NC_4_—insufficient to form channels—and (ii) the small vesicles should have been deflated within the dead (mixing) time of the device if they possessed the alleged water permeability. The DOPC experiments were faulty since Itoh et al. erroneously assumed that the same vesicle could shrink with two different kinetics at once (compare Section 4.1 below). Moreover, Itoh et al. did not reflect on the boundaries physical laws put on water transport through channels. We fill the void by providing theoretical estimates for unitary channel permeabilities based on the diffusional mobility of intraluminal water molecules and calculations of channel access resistance (compare Section 4.2 below).

### 4.1. Criticism of the Stopped-Flow Water Flux Measurements with Fluorous Nanochannels

Itoh et al. [10] reported different scattering traces for the experiments with ^F12^NC_4_ in DOPC vesicles (Figure 5B,C)—yet, neither of them supports fast channel-mediated water transport. The originally published trace does not show any fast deflation kinetics (Figure 5B), indicating sample homogeneity. Accordingly, a single permeability coefficient fits the trace well, whereby *P*_f_ = *P*_f,l_, and thus, *P*_f,c_ = 0. The raw data, distributed several months later, do show both a fast and a slow kinetic (Figure 5C). The genesis of a channel-free vesicle population is unclear, as the provided data indicate an average of 1.9 channels per vesicle (see above) and contain no evidence for an inhomogeneous channel distribution.

However, in the case of 1.9 channels per vesicle, the slow component, which represents the major fraction (Figure 5), should not be observable because the allegedly gigantic channel permeability should have dwarfed the volume flow through the lipid matrix, rendering it negligibly small. Contrary to the approach of Itoh et al. [10], the unitary channel permeability to water and background bilayer permeability cannot both be found from the deflation kinetics of a vesicle ensemble in which each liposome contains superfast water channels (Figure 2) since *P*_f_ = *P*_f,l_ + *P*_f,c_ (Equation (7)) and only *P*_f_ is accessible experimentally. Benevolently, we could assume that the sample was inhomogeneous, allowing for the assignment of the fast component in Figure 5C to water transport facilitated by channels. Yet, this assumption is not warranted as (i) there is no evidence for an inhomogeneous vesicle ensemble and (ii) an increase in nanoring concentration, from which the channels are supposedly built, does not lead to an increase of the fraction of the fast component (Figure 7). Notably, the determination of the fraction of vesicles containing no channel, as well as the number of channels per vesicle in the other fraction, represents a critical challenge in accurate determinations of *p*_f_ and is typically completed in separate experiments [15]. Assumptions on both aforementioned parameters based on the fraction of added compound can be misleading. Finally, it may be added that for calculating water permeability from the time course of osmotically induced volume changes, Itoh et al. used an exponential equation that does not represent a solution of Equation (8).

On the other hand, the experiments with DPPC liposomes were ill-designed. The deflation of channel-containing DPPC vesicles should have occurred within the dead time of the stopped-flow instrument if the channels possessed the alleged water permeability. Inspection of the individual raw fluorescence traces reveals their stochastic behavior and raises the question of what physical process they might reflect—we are confident that it is not rapid water conduction through fluorous nanochannels.

### 4.2. Estimation of the Permeability Limit for Narrow Water-Conducting Channels

Itoh et al. [10] are not the first who claim to have found and successfully measured artificial nanochannels that surpass aquaporins. Narrow single-walled carbon nanotubes are another prominent example [7]. We refuted the claim based on the high activation energy measured for water transport across these tubes [8]. Given the repeated efforts to out-compete millions of years of evolution, it seems warranted to ask whether the water-channeling performance of aquaporins represents the theoretical limit to selective water channels—that is, water channels that are so narrow that hydrated ions cannot penetrate them.

We start by computing water flow, *J*_v_, across a membrane channel from the transmembrane pressure difference:(12)$$J_{v}=L_{p}\left(\Delta P-\Delta \Pi \right)$$
where *L*_p_ is the hydraulic permeability coefficient, Δ*P* the hydraulic pressure difference, and Δ*Π* the osmotic pressure difference across the membrane. We can set Δ*P* = 0 if the experiments are carried out in a liposome suspension, where the channels are located at the vesicular membrane.

At macroscopic scales, Hagen–Poiseuille’s formula successfully relates tube radius, *r*, and channel length, *L*, to *L*_p_, given that *r* << *L*:(13)$$L_{p}=\frac{\pi r^{4}}{8\eta L}$$
where *η* is water viscosity. It is difficult to imagine that Equation (13) works for very narrow channels—be it only because “one shudders to think what a parabolic velocity profile in a single-file pore could mean” [28]. The consensus is that the validity of a continuum description by conventional hydrodynamics (Navier–Stokes equation) breaks down at a length scale of 1 nm. A simple theoretical reasoning is that 1 nm represents the lower bound for defining fluid viscosity, *η* [29]. Experimental support comes, for example, from the breakdown of Poiseuille flow upon confinement of water between graphene oxide sheets spaced < 0.9 nm apart [30]. There is consensus that water transport through narrow channels with a diameter of less than 1 nm is diffusional [28,31,32]. Deviations from Equation (13) observed at the nanoscale are frequently accounted for by abandoning the no-slip condition. Upon introducing a nonzero slip length *b* [33] as *r* approaches tens of nanometer [34], *L*_p_ becomes [29]:(14)$$L_{p}=\frac{\pi r^{4}}{8\eta L}\left(1+\frac{4b}{r}\right)$$

Yet, abandoning the continuum descriptions is not warranted as they are applicable outside the confined space of membrane channels, i.e., they provide information about the access resistance that may limit “ultrafast” water flow. The term access resistance is well known from the study of ion channels, where it signifies diffusional limitations in ion transport to the channel or away from it and the associated concentration changes [35]. The term acquires a new meaning for water channels, as it is not associated with alterations in water concentration close to the channel but may only be realized in terms of limited water mobility. Since *b* can easily reach several hundred nm [34], *L*_p_ could drastically increase if flux was not limited by the access resistance 1/*L*_e_ operating in series with the pore resistance 1/*L*_p_ (Figure 8).

*L*_e_ can be calculated using Sampson’s equation. It is valid for a nanopore of radius *r* and vanishing length [36]:(15)$$L_{e}=\frac{r^{3}}{3\eta }$$
where *L*_e_ is the hydraulic entrance permeability coefficient. *J*_v_ must be the same through the entrance region and through the channel itself. Thus, we should add the hydrodynamic resistances of the entrance region and the channel itself (Equations (14) and (15)). In the limit of *b* >> *r*, the entrance-corrected hydraulic permeability becomes [29]:(16)$$L_{\mathrm{pe}}=\frac{r^{3}}{3\eta }\frac{1}{1+\frac{2L}{\begin{array}{c} 3\pi b \end{array}}}$$

Considering that *b* >> *L* yields:(17)$$L_{\mathrm{pe}}=L_{e}=\frac{r^{3}}{3\eta }$$

That is, the entrance resistance governs channel permeability [36]. Transforming *L*_pe_ (Equation (17)) into *p*_f_ yields:(18)$$p_{f}=\frac{RT\;r^{3}}{3\eta V_{w}}$$

Equation (18) only sets the upper *p*_f_ limit for a cylindrical channel that is as long as the membrane is thick. It specifies the maximum amount of water that can reach the inside of the channel. Besides defining an upper limit, Equation (18) does not allow predictions of the actual amount of water that may pass through such a channel. 

Applying Equation (18), we find *p*_f_ = 1.5 × 10^−13^ cm^3^ s^−1^ for a typical aquaporin with *r* = 0.15 nm [36]. Entrance effects, e.g., an hourglass shape of the vestibule, may augment *p*_f_ [36], so that the theoretical value comes close to the experimentally measured *p*_f_ for GlpF of 1.2 × 10^−12^ cm^3^ s^−1^ [14]. For the artificial nanotube ^F12^NC_4_ of Itoh et al. with *r* = 0.45 nm, we find *p*_f_ = 4.2 × 10^−12^ cm^3^ s^−1^. Thus, the theoretical limit falls short of the reported value of 5.5 × 10^−10^ cm^3^ s^−1^ by a factor of 100.

A second approach at estimating the upper *p*_f_ limit, *p*_limit_, of water movement through the 0.9-nm wide ^F12^NC_4_ nanochannel is based on assuming a superposition of *N*_f_ = 5 water files moving through the pore independently [37] (Figure 5A). This view suggests perfect slip at the channel wall [15,38] and between the water columns. *p*_f,file_ of every water file is intricately linked to the self-diffusion coefficient of liquid water, *D*_w_. To ascertain the upper limit, we assume water retains bulk mobility, i.e., *D*_w_ = 2.57 × 10^−5^ cm^2^ s^−1^ [15,32,39]:(19)$$D_{w}=\frac{p_{f,\mathrm{file}}}{v_{w}}z^{2}$$
where *v*_w_ = 3 × 10^−23^ cm^3^ is the volume of one water molecule and *z* = 0.275 nm is the single-file step distance. A thorough analysis of water transport through numerous biological single-file channels showed that intraluminal diffusivity only approaches but never significantly exceeds its bulk value [9]. The observation also holds in vitro [8] and in silico [40] for single-file carbon nanotubes whose hydrophobic pore walls do not form hydrogen bonds with permeating water molecules. Hence, we may write for *p*_limit_:(20)$$p_{\mathrm{limit}}=N_{f}\frac{D_{w}v_{w}}{z^{2}}=5.1\times 10^{-12}\;{\mathrm{cm}}^{3}s^{-1}$$

The value of *p*_limit_ is surprisingly close to the prediction, *p*_f_ = 4.2 × 10^−12^ cm^3^ s^−1^, made on the basis of Equation (18). Thus, both approaches agree on the fact that the reported *p*_f_ of 5.5 × 10^−10^ cm^3^ s^−1^ for ^F12^NC_4_ cannot be valid.

## 5. Conclusions

Our re-evaluation of the underlying experimental stopped-flow water flux data in [10] demonstrates the failure of ^F12^NC_4_ to facilitate fast water transport. Both the experimental design and data analysis of the original report are flawed. Our analysis of the achievable water permeability limits in narrow channels indicates that aquaporins cannot be significantly out-competed. Limitations in feeding water to the channel and diffusion within the channel both hamper higher permeabilities of selective water channels, i.e., channels that are sufficiently narrow to prevent the flow of hydrated ions.

## Figures and Tables

**Figure 1 biomolecules-13-00431-f001:**
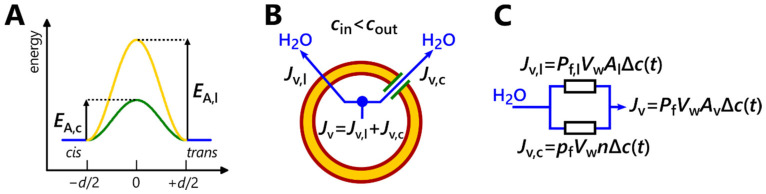
(**A**) The activation energy of transmembrane water transport through water-conducting channels, *E*_A,c_, is lower than that of non-facilitated permeation through the lipid bilayer, *E*_A,l_. *E*_A,c_ is related to unitary channel permeability, *p*_f_, whereby *p*_f_ ∝ exp(−*E*_A_/R*T*) [8,9]. (**B**) In stopped-flow water flux measurements, lipid vesicles are exposed to hyperosmotic conditions by increasing external osmolarity, *c*_out_. Consequently, water exits the vesicle, whereby the volume flux of water, *J*_v_, is assessed by measuring the rate of change of intravesicular volume, d*V*(*t*)/d*t*. Since *V* changes with time, so does internal osmolarity, *c*_in_(*t*), and thus, the osmotic gradient Δ*c*(*t*) = *c*_in_(*t*) − *c*_out_. Water can either traverse the lipid bilayer, *J*_v,l_, or permeate through membrane-embedded water-conducting channels, *J*_v,c_, as indicated by the blue arrows. Since both pathways are available independently, *J*_v_ = *J*_v,l_ + *J*_v,c_. (**C**) An equivalent circuit may illustrate the same: the resistances of the lipid bilayer (1/*P*_f,l_) and that of *n* channels (1/*P*_f,c_) add up, as detailed in Section 2.1. *J*_v,l_ depends on the vesicle surface area covered by lipids, *A*_l_, over which water can permeate. Here, we consider experiments in which the area fraction covered by channels is negligible, i.e., the surface area of the vesicle *A*_v_ is not significantly different from *A*_l_.

**Figure 2 biomolecules-13-00431-f002:**
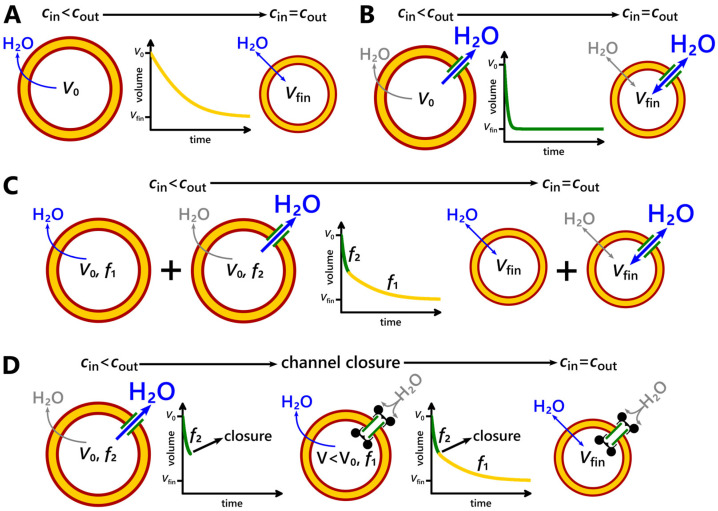
Scheme of vesicular volume changes accessible in a liposome ensemble. Possible water pathways and the composition of the vesicle ensembles differ between the panels. Generally, a population of vesicles with the initial volume *V*_0_ and internal osmolarity *c*_in_ (leftmost vesicles in each panel) is exposed to hyperosmotic conditions by rapidly increasing external osmolarity *c*_out_. The resulting net efflux of water through the available water-conducting pathways (illustrated by unidirectional blue or gray arrows) leads to osmotic equilibrium when *c*_in_ = *c*_out_ (illustrated by bidirectional blue or gray arrows) and *V*_0_ has decreased to *V*_fin_ = *V*_0_ × *c*_0_/*c*_out_, where *c*_0_ is the initial intravesicular osmolarity (rightmost vesicles in each panel). Vesicle volume decreases as *c*_in_ approaches *c*_out_. (**A**) In a homogeneous population of lipid-only vesicles, the lipid bilayer is the only pathway for water conduction, and vesicle volume (yellow curve) decreases with permeability *P*_f,l_ corresponding to that of the bare lipid bilayer. (**B**) In a homogeneous population where each vesicle contains *n* water-conducting channels with a unitary water permeability *p*_f_ far exceeding *P*_f,l_ × *A*_v_, where *A*_v_ is vesicle surface area, the rate of osmotically-driven shrinkage is dominated by water conduction through the channels (green curve). Thus, the total channel permeability *P*_f,c_ = *n* × *p*_f_/*A*_v_ effectively determines the deflation rate. Importantly, there is only a single measurable deflation rate since the vesicle population is homogeneous. (**C**) Given an inhomogeneous vesicle ensemble where 50% are bare lipid vesicles (*f*_1_) and 50% are proteoliposomes containing a single rapidly water-conducting channel (*f*_2_), two different shrinking kinetics are observable. The fast component is dominated by the highly-permeable proteoliposomes (*f*_2_, green curve), which achieve osmotic equilibrium much faster than the bare lipid vesicles (*f*_1_, yellow curve). (**D**) Given a homogeneous ensemble of channel-containing vesicles that nonetheless results in two distinct shrinking kinetics, two interpretations are possible: (i) The channels first facilitate water transport but experience a sudden block (indicated by the Mickey Mouse ears) before vesicle deflation is completed. Consequently, the initial rapid rate of vesicle volume change (*f*_2_) is followed by the rate of non-facilitated water permeation across the bare lipid bilayer (*f*_1_). This scenario appears to reflect the mechanism of water conduction by supramolecular channels that would justify the data treatment performed by Itoh et al. [10] (ii) The fast kinetic has nothing to do with vesicle deflation but indicates some unrelated process (e.g., particle aggregation).

**Figure 3 biomolecules-13-00431-f003:**
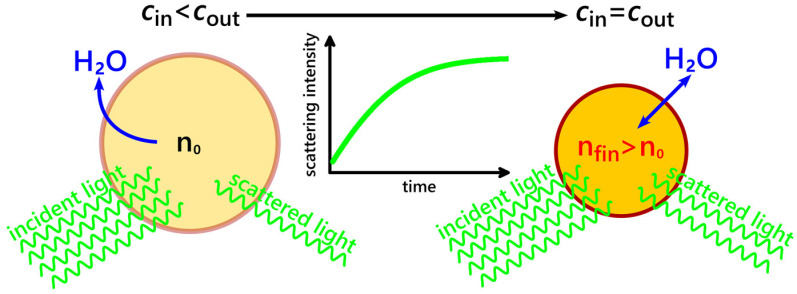
The time course of vesicle volume changes following the introduction of hyperosmotic conditions is accessible via measurements of light scattering intensity *I*(*t*) of the liposome suspension. *I*(*t*) is commonly measured under an angle *θ* of 90°. Although particle size decreases as *c*_in_ approaches *c*_out_ and water leaves the vesicles, *I*(*t*) increases because vesicle deflation increases particle density and, thereby, its refractive index.

**Figure 4 biomolecules-13-00431-f004:**
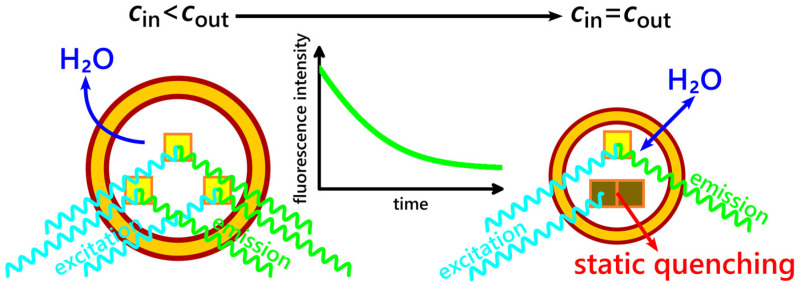
The time course of vesicle volume changes following the introduction of hyperosmotic conditions is accessible via measurements of fluorescence light intensity *I*_f_(*t*) of a suspension of vesicles containing aqueous dyes. Commonly, *I*_f_(*t*) is measured under an angle of 90°. *I*_f_(*t*) decreases because vesicle deflation decreases the distance between the intravesicular fluorophores, and thus, increases their propensity towards self-quenching. For sufficiently high initial fluorophore concentrations (>10 mM for carboxyfluorescein), *I*_f_(*t*) is directly proportional to *V*(*t*).

**Figure 5 biomolecules-13-00431-f005:**
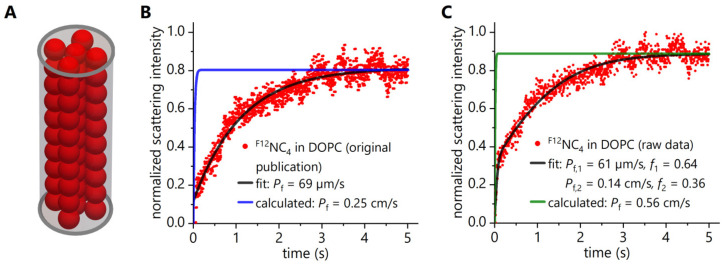
(**A**) Schematic illustration of the hypothetical case in which the 0.9-nm wide ^F12^NC_4_ nanochannel accommodates five independent single-file water columns. (**B**) Analysis of the exemplarily depicted stopped-flow light scattering trace (ref. [10], Figure S38) obtained for DOPC liposomes with diameter *d* = 200 nm (*d* was taken from page 7 of the Suppl. to ref. [10]) in the presence of 0.0053 mol% ^F12^NR_4_. For digitization (red points) we used WebPlotDigitizer. We found vesicle volume *V*(*t*) as a function of time and fitted Equation (9) to *V*(*t*) (black trace), resulting in *P*_f_ = 69 µm/s. As Itoh et al. claimed that ^F12^NC_4_ increases *P*_f_ to ≈0.25 cm/s (read from ref. [10], Figure S38C), we show a calculated curve with this permeability in blue. (**C**) Plotting the raw data (https://doi.org/10.5061/dryad.h18931zq1) reveals two events with different kinetics. At first, we assumed the presence of two vesicle populations (where *f*_1_ and *f*_2_ indicate the relative size of their fractions) with *d* = 176.1 nm (https://doi.org/10.5061/dryad.h18931zq1) possessing permeabilities *P*_f,1_ and *P*_f,2_. The inset states the results of the fitting procedure [14], revealing a major fraction of empty DOPC vesicles. If each vesicle contains at least one channel with *p*_f_ = 5.5 × 10^−10^ cm^3^ s^−1^, deflation kinetics resembling the green line should be expected. The genesis of the minor fraction with fast kinetic is unclear. Itoh et al. assigned the fast component to *P*_f,c_ and the slow component to *P*_f,l_. Taken together, the authors claim a scenario like in Figure 2D where Mickey Mouse ears miraculously plug the channel after only a few ms. Such a scenario is implausible, since ^F12^NC_4_ does not possess an intrinsic blocking mechanism.

**Figure 6 biomolecules-13-00431-f006:**
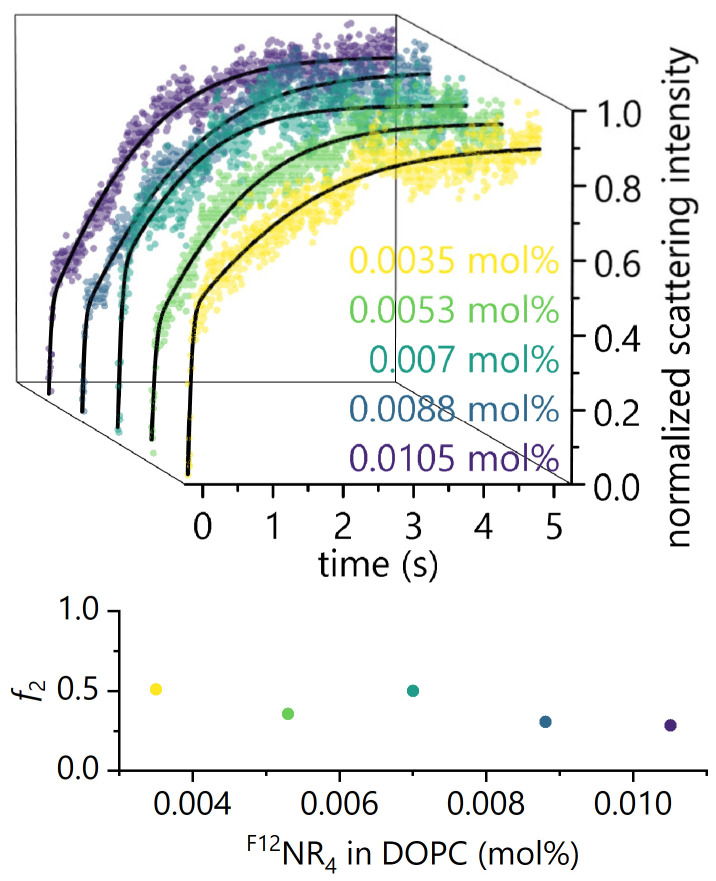
Paradoxically, increasing the amount of ^F12^NR_4_ nanorings added to DOPC appears to increase the fraction, *f*_1_, deflating with the kinetics of the bare lipid bilayer, i.e., more channel components added yielded fewer channel-containing vesicles. The stopped-flow light scattering traces of DOPC vesicles with increasing fractions of ^F12^NR_4_ nanorings were averaged and normalized, as suggested by Itoh et al. [10] (https://doi.org/10.5061/dryad.h18931zq1). As the main text outlines, the scattering traces are incompatible with a homogeneous ensemble of vesicles, each containing ^F12^NC_4_ channels with the alleged extreme water permeability (see Figure 2). To that effect, we fit the traces assuming two vesicle populations [14]: one without channels indicating background lipid permeability (*f*_1_) and one with faster kinetics (*f*_2_). The fit reports the magnitude of both fractions. The relative contribution of the fast population (*f*_2_) is plotted against the fraction of ^F12^NR_4_ nanorings added to DOPC. We find no positive correlation between *f*_2_ and the amount of added ^F12^NR_4_ nanorings. This means that increasing the amount of ^F12^NR_4_ added to lipids does not increase the fraction of vesicles containing functional ^F12^NC_4_ channels.

**Figure 7 biomolecules-13-00431-f007:**
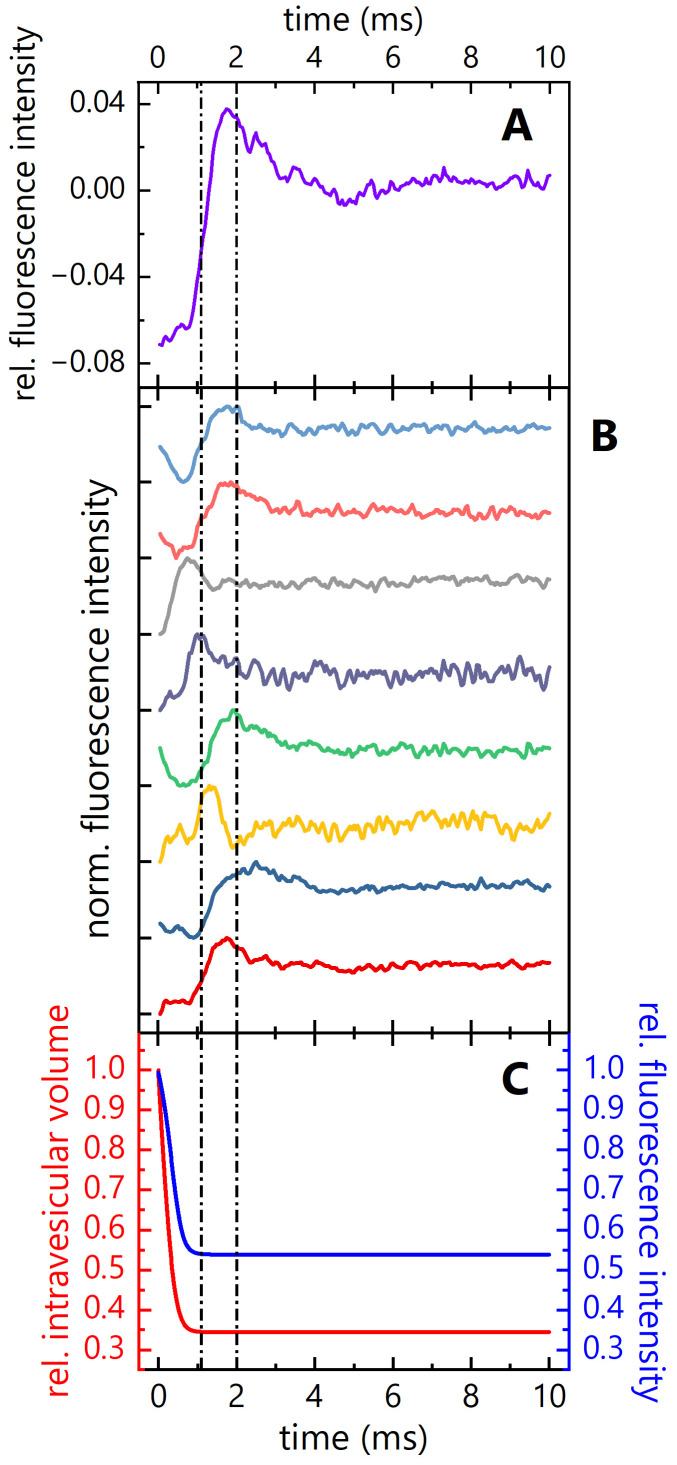
A comparison between fluorescence intensity traces obtained in stopped-flow self-quenching experiments on DPPC liposomes comprising 0.0018 mol% ^F12^NR_4_ nanorings [10] and calculated data based on the reported permeability of ^F12^NC_4_ in DPPC. (**A**) Shown is the exemplary trace published as Figure 3B in ref. [10] in its entirety. Data evaluation is not possible during the dead time of the stopped-flow instrument, which Itoh et al. [10] specified as 1.1 ms—illustrated by the first dash-dotted vertical line. In contrast to the expected decrease, fluorescence intensity after 1.1 ms is still increasing. Yet, the authors chose to display the exemplary trace only starting from 2 ms (illustrated by the second dash-dotted line), where it shows the expected decline in fluorescence intensity. (**B**) The constituent traces from which Itoh et al. generated the exemplary trace by averaging and subsequent normalization. The displayed curves are normalized to the interval [0, 1]. After 1.1 ms, most traces are still climbing; even after 2 ms, the individual traces do not behave uniformly. (**C**) The time course of intravesicular volume was calculated using Equation (9) and is plotted as relative volume over time (red trace). The following parameters were used for calculation: *c*_0_ = 210 mM, *c*_out_ = 610 mM (as published and assuming an osmotic coefficient of 1), *P*_f_ = 0.2 cm/s (read from Figure 3C in Itoh et al.), *r*_0_ = 23.6 nm (read from the communicated raw data). To allow for comparison with panels A and B, relative intravesicular volume was transformed into relative fluorescence intensity as described by Itoh et al. [10] in a document accompanying the communicated raw data (blue trace). Since the authors chose to use the impractically low initial intravesicular carboxyfluorescein concentration of 0.5 mM, the transformation between fluorescence intensity and volume is not linear. The abscissa is identical in each panel.

**Figure 8 biomolecules-13-00431-f008:**
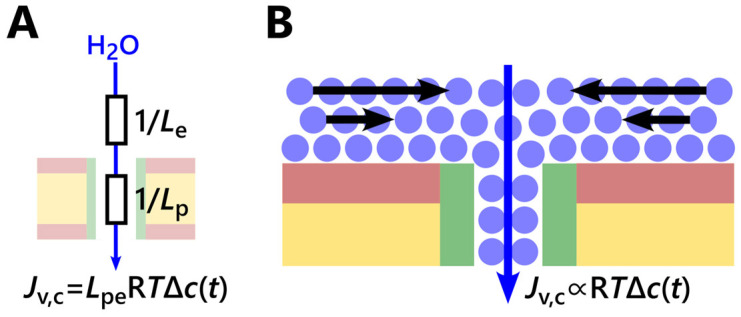
(**A**) Entrance effects may limit rapid water transport through selective water channels. The hydrodynamic resistance of the entrance region, 1/*L*_e_, determines the rate at which water can be fed to the channel. 1/*L*_e_ is in series with the hydrodynamic resistance of the channel, 1/*L*_p_, which is intrinsic to the channel and governed by interactions of intraluminal water molecules with the pore walls [14,34]. If *L*_p_ exceeds *L*_e_, the water flux through the narrow channel, *J*_v,c_, is limited by *L*_e_ (Equation (17)). (**B**) One way of rationalizing the access resistance for water is to consider the no-slip condition at the membrane–water interface. The further into the bulk, the more mobile the water molecules become, as indicated by the horizontal arrows. At the pore mouth, highly mobile water molecules are effectively depleted due to the osmotic gradient across the membrane, where R*T*Δ*c*(*t*) is osmotic pressure across the vesicular membrane. Since (a) permeating water molecules have to be replenished from the surrounding and (b) interfacial water immediately adjacent to the pore mouth is slowed down, *L*_e_ must depend on pore radius.

## Data Availability

Not applicable.

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
