# Peer review of "Tutorial for Stopped-Flow Water Flux Measurements: Why a Report about “Ultrafast Water Permeation through Nanochannels with a Densely Fluorous Interior Surface” Is Flawed"

_biomolecules, 2023, doi:10.3390/biom13030431_

Round 1

Reviewer 1 Report

The main objective of the present study was to monitor water channeling performance of the fastest aquaporins. The findings were quite useful to facilitate fast water transport. Although this basic study is useful for the readers but there are several issues that are of major concern. Therefore the manuscript needs rigorous revision before publishing.

Abstract:

Abstract should have more focus on key findings and potential future application.

Introduction:

Comment
-I suggest to elaborate this section. Note that this section should be well structured, and inclusive of all information to the development of your findings.

Material methods:

This section is well described.

Results:

The result section should be more concise.

Try to be specific and focus on your key findings or results.

The authors should exchange to more clear pictures.

Discussion:

The discussion is too long with numerous (re)writing of ideas

Elaborate the results comparing with the earlier studies.

The discussion needs to be much more compact while some results are missing explanations, needs to be rectified.

This section would benefit from some careful revision. I suggest you minimize similar/same style of discussing/ reporting.

Author Response

See attachement.

Reviewer 2 Report

The study of the water transport through sub-1-nm channels is important for the biosystem, nanofluidic, and membrane technology. This work insightfully questioned the reported work in Science and provided sufficient results to prove the flawed report. This work is interesting and is recommended to be accepted by the Journal of Biomolecules. 

Author Response

Below, the reviewers’ comments are reproduced with our replies interspersed in [[brackets in blue type]].

The study of the water transport through sub-1-nm channels is important for the biosystem, nanofluidic, and membrane technology. This work insightfully questioned the reported work in Science and provided sufficient results to prove the flawed report. This work is interesting and is recommended to be accepted by the Journal of Biomolecules.

[[We thank the reviewer for the positive assessment! ]]

Reviewer 3 Report

The paper “Tutorial for stopped-flow water flux measurements or why a recent Science report about ultrafast fluorous nanochannels is flawed” by Pfeffermann and Pohl reports an interesting reevaluation of the results, published previously in Science Magazine by Y. Itoh et.al in “Ultrafast water permeation through nanochannels with a densely fluorous interior surface”. As soon as the publication of sensational issues with further refuses on providing disclaimers became a common practice of high-reputed journals, I greatly appreciate and support the efforts of Pfeffermann and Pohl. However, detailed analysis stopped me from suggesting the paper to be published in Biomolecules as is and insist on major revision.

The major concern is connected to evaluation of water permeability (permeance) of the channel by both Pfeffermann and Pohl and the authors of original paper. In the present notation, the volumetric water flux defined in equations 3 and 4 represent the term APfVw considering Pf= P=+Pf,c, i.e. summing directly the permeabilities (permeances) of the membrane and the channel. On the other hand, Pf has the dimension of [cm/s] or [cm3/cm2/s], i.e. being an areal value. Only fluxes, but not the permeabilities are additive for parallel flows. As the surface area of vesicle is very different from the surface area of the channel, these evaluations may be erroneous in the present form. Therefore, the authors are strongly recommended to consider the surface areas in the evaluations and provide a direct comparison and discussion of area values used in their reevaluation treatment with those reported by Y. Itoh et.al. The estimate for the role of errors in those areas is also greatly appreciated.

Few minor remarks are given below:

1.    Title: Probably the authors may modify the title of the manuscript to make it searchable with standard quick search engines like Google, Bing, etc together with the original paper. To my opinion, inclusion of the title of original paper is necessary in order to warn the researchers on the potential disclaimer, which was refused from publishing by the original journal.

2.    P2, Line 46: The authors state “Lowering Pf commonly accompanies a reduction in EA”. Misprint?

3.    P9-10, Lines 370-385: The authors are recommended to keep away from including Hagen-Pouseille equation (even with slip-flow terms) for sub-nanometer channels in order to avoid confusing the Readers. Despite water flow in the channel of R=0,45 nm can sometimes still be considered as continuous medium, (as the authors mention at line 426), that is counterintuitive. To the best of my experimental knowledge, the edge for bulk water transport in 2D channel is indeed 0.9 nm (see fig 3f in https://iopscience.iop.org/article/10.1088/2053-1583/ab15ec). However, one can foresee the edge will be higher for 1D channels. I see no need for giving eqn. 8-9 in the text.

PS: The comment on the possibility of fast transport in carbon nanotubes given in the paper is insightful. In order to provide you with some experimental evidences for slow water and ions transfer kinetics inside nanotubes, you can look on high-resolution TEMs [https://pubs.acs.org/doi/10.1021/nl0484907; https://pubs.acs.org/doi/full/10.1021/acs.jpcc.7b06100]. A typical timeframe for the image acquisition is 0,1-1 s, which is a huge period of time, when no movement of water or ions occur. The resulting diffusion coefficients are much lower compared to any of the reported values.

PPS: None of the involved references are provided for citation, but are only given to the Authors as external estimates for their results.

Round 2

Reviewer 3 Report

The Authors have partly alleviated my concerns, introducing the corresponding corrections to the manuscript text. I still feel some doubt on applicability of substitution of single channel permeability Pf with the areal property of Pf,c given by eqn.5, as it also substitutes the area of the channel by the whole vesicle area. To my opinion, this can lead to the discrepancy in the permeability, derived by Itoh et.al and the Authors. Nevertheless, the Authors have now provided all the calculation details and adequate explanations for readers. The reported viewpoint is anyway important and insightful. Therefore, I recommend the paper for publication, leaving the issue at the Authors disposal and conscience.